# Immunomodulatory Effects of Probiotics on COVID-19 Infection by Targeting the Gut–Lung Axis Microbial Cross-Talk

**DOI:** 10.3390/microorganisms10091764

**Published:** 2022-08-31

**Authors:** Kalliopi D. Synodinou, Maroulla D. Nikolaki, Konstantinos Triantafyllou, Arezina N. Kasti

**Affiliations:** 1Department of Nutrition and Dietetics, Attikon University General Hospital, 12462 Athens, Greece; 2Hepatogastroenterology Unit, 2nd Department of Propaedeutic Internal Medicine, Medical School, Attikon University General Hospital, National and Kapodistrian University, 12462 Athens, Greece

**Keywords:** SARS-CoV-2, gut–lung axis, immunomodulation, probiotics, bacteriocins

## Abstract

The ecosystem of the human gastrointestinal tract, named gut microbiota, represents the most thoroughly mapped ecosystem. Perturbations on bacterial populations cause dysbiosis, a condition correlated to a wide range of autoimmune, neurological, metabolic, cardiovascular, and respiratory diseases. The lungs have their flora, which are directly related to the gut flora via bidirectional communication allowing the transport of microbial metabolites and toxins produced by intestinal bacteria through the circulation and lymphatic system. This mutual microbial cross-talk communication called the gut–lung axis modulates the immune and inflammatory response to infections. COVID-19 causes dysbiosis, altered intestinal permeability, and bacterial translocation. Dysbiosis, through the gut–lung axis, promotes hyper-inflammation, exacerbates lung damage, and worsens clinical outcomes. Preclinical and clinical studies have shown that probiotics can regulate cytokine secretion, thus affecting both nonspecific and specific immunity. Probiotics act by blocking the virus from invading and proliferating in host cells, by stimulating the immune response, and by suppressing the activation of NLRP3 inflammasome. Herein, we reviewed the evidence from preclinical and clinical studies evaluating the effect of probiotics administration on the immune response to COVID-19 infection by targeting the gut–lung axis microbial cross-talk.

## 1. Introduction

The gastrointestinal tract is colonized by a wide range of dynamic microbial populations, consisting mainly of anaerobic bacteria, known as gut microbiota. Direct microbial–host interaction affects many physiological functions such as digestion, metabolism, intestinal barrier integrity (epithelial barrier), organ function, and immune homeostasis [1]. Bacteria usually live in a symbiotic relationship with the host, while perturbations in their composition and function cause dysbiosis associated with gastrointestinal, neurological, metabolic, cardiovascular, and respiratory disorders [2]. Lungs, for years, have been considered to be sterile because microbiological culture techniques have shown negative results. Nowadays, technological advances confirm that healthy lungs host a dynamic and diverse bacterial ecosystem [3]. The predominant genera that colonize healthy lungs are Bacteroidetes and Firmicutes, followed by Proteobacteria and Actinobacteria [4]. Dysbiosis can weaken nonspecific lung immunity, promote the host’s susceptibility to infections, and may worsen chronic lung disease. In addition, changes in lung flora affect cytokine production by triggering the transition from anti-inflammatory to proinflammatory cytokines, thus promoting injury to the pulmonary epithelium resulting in fibrosis [4]. The interaction between lung and gut microbiota involves a bidirectional communication with metabolites and toxins produced by intestinal bacteria arriving at the lungs through the circulation and lymphatic system, known as the “gut-lung axis” [5]. Microbiota-immune interactions regulate the inflammatory response by activating this axis through bacterial populations and their by-products which leak the mucosal barrier (translocation) [2]. The bacterial translocation, leukocyte transport, and inflammatory cytokine secretion through the gut–lung axis are phenomena also found in SARS-CoV-2 infection, worsening the clinical outcome of patients [6]. Although COVID-19 is primarily a viral respiratory disease caused by SARS-CoV-2, more and more evidence suggests that the gastrointestinal tract is involved in COVID-19 [7]. SARS-CoV-2 infects intestinal epithelial cells (IECs) too, due to its transport through the lymphatic system, resulting in disruption of intestinal permeability and mucosal integrity [8]. Thus, there appears to be a vital cross-communication between the microbial communities in the mucous membranes of the lungs and the intestine [4]. Probiotics have well-documented antiviral activities against common respiratory viruses such as respiratory syncytial virus, influenza, and rhinovirus. Probiotics regulate cytokine secretion, thus affecting both nonspecific and specific immune response. Bacteriocins, produced by lactic acid bacteria (LAB), are proteinaceous substances with antimicrobial properties that can inhibit the adhesion and invasion of other agents in the intestinal epithelium [9], as shown in Figure 1.

### 1.1. Preclinical Trials Supporting the Gut–Lung Axis Communication in COVID-19

Sencio et al. used Syrian hamsters for their exceptional ability to mimic the onset of pathological manifestations of COVID-19 disease. The infected animals had an abundance of opportunistic pathogenic bacteria, such as *Enterobacteriaceae* and *Desulfovibrionaceae*, a decrease in beneficial Short-Chain Fatty Acids (SCFAs)-producing *Ruminococcaceae* and *Lachnospiraceae,* and reduced concentration of SCFAs in the blood. Since SCFAs have a dual function, namely, a protective role in maintaining the integrity of intestinal epithelial barrier and regulating inflammatory response, the above findings were considered compatible with dysbiosis. However, SCFAs supplements failed to improve the clinical outcomes [13].

Studying primates, Sokol et al. infected macaques with SARS-CoV-2 and observed alterations in the gut flora composition, which peaked on the 13th day postinfection. The relative abundance of the genus *Acinetobacter* (Proteobacteria) and the family *Ruminococcaceae* (Firmicutes) were associated with viral presence in the upper respiratory tract, and *Peptostreptococcaceae* (especially of the Intestinibacter) was so in the rectum. These bacterial strains were associated with increased inflammatory markers in blood samples (C-reactive protein and proinflammatory cytokines). Additionally, a positive correlation was identified among several species of the genus *Streptococcus* and chemokine production after viral infection. On the other hand, SARS-CoV-2 reduced SCFA levels in macaque feces, which affected the metabolism of tryptophan, a precursor of serotonin, and significantly altered the synthesis of various metabolites and bile acids. It remains unclear if the decrease in SCFAs concentrations may be related to lower production from gut bacteria or higher needs from the host cells. Researchers concluded that the reduced SCFA levels may be associated with proinflammatory conditions, increasing susceptibility to viral respiratory infections, including COVID-19, while the infected macaques developed mild symptoms of pneumonia, similar to those in people with COVID-19 [1].

Groves et al. evaluated the effect of a viral lung infection on gut flora composition by replicating infection with common respiratory pathogens such as respiratory syncytial virus (RSV) and influenza virus in mice. Animals were inoculated with a vaccine strain of live attenuated influenza virus (LAIV). Results after inoculation showed that RSV and influenza infection led to significant changes in the diversity of the intestinal flora, with an increase in Bacteroidetes phylum and a concomitant decrease in the abundance of Firmicutes. Although Bacteroidetes growth was consistently repeatable across multiple experiments, differences were observed in the bacterial family and taxonomy level. This fact suggests a change in the gut environment following a viral respiratory infection, which favors the growth of the general Bacteroidetes population, but not individual bacterial families. No change in gut flora composition was observed after vaccination with live attenuated influenza vaccine (LAIV), suggesting that the cause of gut microbiota change is related to the virulence of the viral strain. Viral respiratory infections also led to increased lipocalin-2 (a biomarker of gut inflammation) in stool, indicating low-grade gut inflammation and increased levels of Muc5ac mucin in both the airway mucosa and the colon. This study highlights the gut microbiota alterations after viral respiratory infections, whereas these changes were not observed during vaccination. Whether increased mucus levels and gut inflammation are the cause or effect of these changes is not yet clear [14].

### 1.2. Clinical Trials Supporting the Gut–Lung Axis Communication in COVID-19

Although COVID-19 is primarily considered a respiratory infection, there is growing evidence supporting the GI involvement in the disease. Jin et al. in China analyzed 74 (11.4%) of 651 patients with COVID-19 who had at least one gastrointestinal (GI) tract symptom (nausea, vomiting, and diarrhea). Diarrhea was the most common symptom, accounting for 8.14% of the enrolled patients. Jin et al. reported, for the first time, novel characteristics of SARS-CoV-2 and its ability to infect the GI tract [15].

In Hong Kong, fecal samples from 15 patients with COVID-19 were analyzed and microbiome data were compared with those from 6 subjects with community-acquired pneumonia and 15 healthy individuals (controls). Patients with COVID-19 had significant alterations in gut microbiota compared to controls, characterized by an abundance of opportunistic pathogens and a reduction of beneficial microbes during hospitalization. Gut dysbiosis persisted even after SARS-CoV-2 and the resolution of respiratory symptoms. The baseline abundance of *Coprobacillus*, *Clostridium ramosum*, and *Clostridium hathewayi* correlated with COVID-19 severity. There was an inverse correlation between the higher populations of *Faecalibacterium prausnitzii* (an anti-inflammatory bacterium) and disease severity. During the hospitalization, *Bacteroides dorei*, *Bacteroides thetaiotaomicron*, *Bacteroides massiliensis*, and *Bacteroides ovatus*, which reduce ACE2 expression in the murine gut, were correlated inversely with SARS-CoV-2 load in patients’ fecal samples [16].

Tao Zuo et al. collected fecal samples from patients with COVID-19 during hospitalization until discharge. Subsequently, they compared fecal mycobiome compositions of 30 patients with those of 9 subjects with community-acquired pneumonia and 30 healthy individuals (controls) and assessed their profiles during hospitalization until viral clearance. Samples from patients with COVID-19 were abundant in opportunistic pathogens, such as *Candida albicans*, *Candida auris*, and *Aspergillus flavus*, compared to controls. *A. flavus* and *Aspergillus niger*, two respiratory-associated fungal pathogens, were detected in fecal samples from a subgroup of patients with COVID-19, even after clearance of SARS-CoV-2 from nasopharyngeal samples and resolution of respiratory symptoms supporting the gut–lung axis hypothesis [17]. In another pilot study, the same research group provided evidence from 7 of 15 patients with COVID-19 who had stool positivity for SARS-CoV-2 by viral RNA metagenomic sequencing even in the absence of GI manifestations. Patients had active and prolonged quiescent GI infection even after recovery from respiratory symptoms of SARS-CoV-2. The gut microbiota of these patients were characterized by enrichment of opportunistic pathogens *Collinsella aerofaciens*, *Collinsella tanakaei*, *Streptococcus infantis*, *Morganella morganii*, and loss of beneficial bacteria [18].

Yeoh et al. collected and analyzed blood samples from 100 patients and stool samples from 27 of the total patients up to 30 days after clearance of SARS-CoV-2. Then, they measured inflammatory cytokines and blood biomarkers in blood plasma. Results showed increased IL-10, TNF-α, and biomarkers such as C reactive protein, lactate dehydrogenase, aspartate aminotransferase, and gamma-glutamyl transferase. Gut flora were significantly altered in patients with COVID-19 compared to healthy controls. Bacteria with immunomodulatory activity, such as *Faecalibacterium prausnitzii*, *Eubacterium rectale*, and Bifidobactria, were diminished and remained reduced 30 days after recovery. Alterations in gut microbiota were associated with disease severity [19].

Li et al. examined the gut microbiota of 47 patients with COVID-19 and compared them with those of healthy controls (n = 19). Fecal samples showed four bacterial species, *Streptococcus thermophilus*, *Bacteroides oleiciplenus*, *Fusobacterium ulcerans*, and *Prevotella bivia.* Bacterial strains of *Bacteroides stercoris*, *B. vulgatus*, *B. massiliensis*, *Bifidobacterium longum*, *Streptococcus thermophilus*, *Lachnospiraceae bacterium* 5163FAA, *Prevotella bivia*, *Erysipelotrichaceae bacterium Erysipelotrichaceae* 6145, and *Erysipelotrichaceae* 2244A were found in abundance in COVID-19 patients, while *Clostridium nexile*, *Streptococcus salivarius*, *Coprococcus catus*, *Eubacterium hallii*, *Enterobacter aerogenes*, and *Adlercreutzia equolifaciens* were reduced. The relative abundance of butyrate-producing *Roseburia inulinivorans* was markedly reduced in COVID-19 patients, while *Paraprevotella* sp. and *Streptococcus thermophilus* were increased. This study showed that *Roseburia inulinivorans*, *Bacteroides faecis*, *Bifidobacterium bifidum*, *Parabacteroides goldsteinii*, *Lachnospiraceae bacterium* 9143BFAA, and *Megasphaera* sp. Had no correlation with disease severity, while *Paraprevotella* sp., *Streptococcus thermophilus*, *Clostridium ramosum*, and *Bifidobacterium animalis* were positively correlated with the severity of COVID-19 [20].

In another study, Vestad et al. collected blood plasma from patients on hospital admission and three months after severe COVID-19 for the assessment of inflammatory markers and intestinal barrier dysfunction. During the follow-up, pulmonary function was assessed by measuring the diffusing capacity of the lungs for carbon monoxide (DL_CO_). Rectal swabs were collected for gut microbiota analysis by sequencing the 16S rRNA gene. Results showed that gut flora decreased in COVID-19 patients with respiratory dysfunction (defined as the ability to DL_CO_ below the threshold three months after hospitalization). Specifically, decreased abundance of the family *Erysipelotrichaceae* UCG-003 and increased *Flavonifractor* and *Veillonella* species were observed, with *Veillonella* being possibly associated with fibrosis of the lung parenchyma. During hospitalization, elevated levels of lipopolysaccharide-binding protein (LBP) in patients’ blood plasma were directly correlated with respiratory failure, defined as pO_2_/fiO_2_ < 26.6 kPa. LBP levels remained elevated during and after three months of hospitalization and were associated with low-grade inflammation and respiratory dysfunction. Respiratory dysfunction after COVID-19 was associated with disturbances in gut flora, which enhance the gut–lung axis hypothesis [7].

## 2. Literature Search Strategy

In this review, we aimed to explore the effect of probiotics on the immune response to COVID-19 infection by targeting the gut–lung axis microbial cross-talk. We assessed preclinical and clinical studies focusing on probiotic intake by subjects infected with SARS-CoV-2 and their effects on the gut–lung axis. Literature research using the terms “SARS-CoV-2 & gut-lung axis”, “gut bacteria & COVID-19”, “Coronavirus & gut-lung axis & ACE2”, “probiotics & cytokine storm & COVID-19”, “oral bacteriotherapy”, “COVID-19 & immunobiotics”, and their combination in PubMed revealed eight publications in the English language (accessed on 30 May 2022). We used evidence from original articles only and we included studies identified by manual search of the reference lists of the aforementioned articles. Reviews, abstracts, conference presentations, editorials, and study protocols were excluded. Figure 2.

## 3. Preclinical Studies of Probiotic Administration in COVID-19 Infection

Pham et al. experiments focused on intranasal inoculation of mice with M glycoprotein, one of the major glycoproteins of SARS-CoV-2, or with a green fluorescent protein (GFP). Animals infected with M protein showed a significant increase in interleukin IL-6 in bronchoalveolar lavage fluid (BALF) compared to the GFP group. High levels of M-protein-induced IL-6 appeared to be significantly reduced in phosphodiesterase 4 (PDE4B) knockout mice. Although PDE4B plays an essential role in the secretion of IL-6, there was no difference in IL-6 levels in transgenic mice in the absence of PDE4B, either inoculated intravenously with recombinant M protein or with GFP. The addition of mycelia to the diet caused an increase in *L. rhamnosus* in their gut flora. Coadministration of *L. rhamnosus* with mycelia for two weeks significantly reduced IL-6 and M protein-induced PDE4B expression. The probiotic action of *L. rhamnosus* with mycelia against M-protein was reversed in mice treated with GLPG0974, a potent free fatty acid 2 (FFAr2) receptor antagonist. FFAr2 has high affinity for butyric acid, and is expressed by a variety of cells, including macrophages and epithelial cells of nasal cavities and lungs. SCFAs can bind to FFAr2, thereby reducing inflammatory processes, thus highlighting the essential role of FFAr2 for the probiotic activity of *L. rhamnosus* EH8 against the infectious activity of SARS-CoV-2 M-protein. Understanding the host–microbial interaction along the gut–lung axis highlights the crucial role of SCFAs in stimulating specific and nonspecific immune responses in the bone marrow to treat airway inflammation. Data have shown that PDE4B has a key role that contributes to the M-protein-induced secretion of IL-6. Coadministration of *L. rhamnosus* EH8 with mycelia is beneficial, as it mediates counteraction with the inflammatory effect of protein M through FFAr2 of SARS-CoV-2 [21].

*L. plantarum* GUANKE (LPG) in mice promoted specific SARS-CoV-2 immune responses in both inflammatory cascade and induction of innate immune memory through enhancing the interferon signaling pathway and suppressing inflammation and cell apoptosis. LPG enhanced humoral immune response against SARS-CoV-2 and stabilized the T-cell receptor-binding domain (RBD)-specific stimulation in mice when administered immediately after inoculation with SARS-CoV-2. Oral administration of LPG produced neutralizing antibodies even six months after immunization. The immediate administration after vaccination enforced the production of neutralizing antibodies, which was eight times higher in bronchoalveolar lavage fluid (BALF) and doubled in blood serum. T-lymphocyte activity was both stable and prolonged in BALF, as well as in spleen. Transcriptional analysis revealed that LPG limits inflammatory response, cell apoptosis, and triggers interferons (IFNs) activation without prior vaccination. Stimulation of IFNs could promote specific immune responses. Oral LPG administration enhances gene expression associated with T- and B-cell activation, proliferation, survival, differentiation into memory cells, and migration to tissues. Therefore, if LPG is coadministered with the vaccine, it will enhance SARS-CoV-2-specific T- and B-lymphocyte responses, facilitating their mobilization in the mucosa and inducing the creation of cells’ immune memory. In addition, LPG can affect gut–spleen and gut–lung immune regulatory axes by crossing the mucosal intestinal barrier through systematic mobilization of the immune responses it induces. These results suggest that LPG strain could be coadministered with the vaccine acting as an adjuvant to induce specific and nonspecific immunity, thus enhancing immunization and extending vaccination protection for SARS-CoV-2 [22] Figure 3.

## 4. Clinical Trials with Probiotic Administration in COVID-19 Infection

Wu et al. investigated the volatile and heterogeneous gut microbiota shifts of COVID-19 patients over the course of a probiotics-assisted therapy. Three groups participated in this study: the study group was COVID-19 patients, the control group was patients with community-acquired pneumonia (CAP), and healthy controls. Patients with either COVID-19 or CAP were treated with antivirals, antibiotics, thymosin (hormone), vitamin C, and ulinastatin, while the study group also received probiotics with nine different Lactobacilli strains daily. COVID-19 patients showed significant alterations in their gut microbiota characterized by great heterogeneity from patient to patient, but also from the time of taking the sample. Alterations were observed in the respiratory flora as well. Taxonomic changes showed a reduction of *Faecalibacterium*, *Roseburia*, and *Clostridium* XlVa strains and increased proportions of *Enterococcus*, *Rhodococcus*, and *Acinetobacter*. Transcriptional alterations in gut microbiota included increased opportunistic and nonopportunistic pathogens (e.g., *Escherichia coli*, *Salmonella enterica*, *Staphylococcus auricularis*, and *Klebsiella pneumoniae*) and antibiotic resistance genes, as well as a reduction of the symbiotic butyric acid-producing *Faecalibacterium prausnitzii*. After treatment, pulmonary dysfunction was partially restored, as well as, was intestinal dysbiosis (detected by in-creased microbial diversity) in COVID-19 patients. Probiotic-assisted therapy reduced inflammatory markers such as TNF-α, IL-1β, IL-4, and IL-12P70 [23].

Gutiérrez-Castrellón et al. conducted a single-center, quadruple-blinded, randomized trial in adults with COVID-19. *L. plantarum* (KABP022, KABP023, KABP033) and *Pediococcus acidilactici* KABP021 or placebo were administered in Intensive Care Unit (ICU) patients. Researchers measured (i) the proportion of patients in remission on day 30, (ii) the proportion of hospitalized patients with moderate or severe disease or deaths, and (iii) days in the ICU. Complete remission was achieved in 53.1% and 28.1% in the probiotics and placebo groups, respectively. Probiotics reduced the viral load in the nasopharyngeal cavity, reduced pulmonary infiltrates, and the duration of digestive and nondigestive symptoms compared with placebo. No significant alteration was observed in gut microbiota between groups. Probiotic administration significantly increased the production of specific IgM and IgG responses against SARS-CoV-2 compared to placebo. D-dimer levels decreased in both groups, but probiotic activity decreased more rapidly in the intervention group compared to placebo. High D-dimer levels were associated with increased risk of incident venous thromboembolism, such as pulmonary embolism, and were associated with the severity and mortality of COVID-19. Therefore, probiotics acted beneficially mainly via their interaction with the host immune system rather than through the composition of the gut flora [24].

D’Ettorre et al. evaluated the role of oral bacteriotherapy with *Streptococcus thermophilus* DSM 32345, *L. acidophilus* DSM 32241, *L. helveticus* DSM 32242, *L. paracasei* DSM 32243, *L. plantarum* DSM 32244, *L. brevis* DSM 27961, *B. lactis* DSM 32246, and *B. lactis* DSM 32247 as a supplementary therapeutic strategy in COVID-19 disease. Seventy hospitalized patients with COVID-19 fever, and a Computed Tomography with detected lung involvement of more than 50% who required noninvasive oxygen therapy, were included in the study. Forty-two patients were treated with hydroxychloroquine, antibiotics, and monoclonal antibodies (tocilizumab), either alone or in combination. The second group of twenty-eight patients received the same treatment, to which oral bacteriotherapy was added. Almost all patients receiving bacteriotherapy experienced remission of diarrhea and other symptoms within 72 h compared with less than half of the controls. Although not statistically significant, higher rates of ICU admission and mortality occurred in the control group and the estimated risk of developing respiratory failure was eight times lower in the group with oral bacteriotherapy [5].

The elderly are more susceptible to SARS-CoV-2 infection and display weakened immune responses to COVID-19 vaccine. Probiotic strain *Loigolactobacillus coryniformis* K8 CECT 5711 administration was evaluated in people over 60 years of age on the SARS-CoV-2 vaccine-induced immune responses. Two hundred volunteers were randomly allocated to receive *L. coryniformis* K8 strain or a placebo daily for three months. All participants completed mRNA vaccination, while the intervention started ten days after the first dose. Specific IgA and IgG antibody levels were measured 56 days after the completion of vaccination, showing that there were no differences in antibody levels between groups. Nineteen subjects were infected with COVID-19 during the intervention (eleven received *L. coryniformis* K8 and eight placebo). The results revealed that IgG levels in the *L. coryniformis* K8 group were significantly higher, though at ages 85 and older, probiotic administration increased IgA antibody levels [25] Table 1.

## 5. In Vitro Trials with Probiotic Challenge in Cell Lines

Islam et al. evaluated the immunomodulatory effects of viable and nonviable *Lactiplantibacillus plantarum* strains in human respiratory epithelial cells (Calu-3) and assessed their potency to reduce the susceptibility of Calu-3 to the acute respiratory syndrome from SARS-CoV-2 infection. *L. plantarum* MPL16 and CRL1506 regulated the expression of proinflammatory cytokines and chemokines. Researchers demonstrated that early administration of *L. plantarum* strains to Calu-3 significantly increased the production of the cytokines IFN-β and IL-6, while they decreased the chemokines CCL5, CXCL8, and CXCL-10. In addition, the strains MPL16 and CRL1506 increased the resistance of Calu-3 to SARS-CoV-2 inoculation test. Furthermore, *L. plantarum* MPL16 was more effective than the strain CRL1506. It is worth noticing that neither the nonviable strains MPL16 and CRL1506 nor the nonimmunomodulatory strains of *L. plantarum* CRL1905 and MPL18 could change the resistance of Calu-3 to SARS-CoV-2 infection. These data enhance the evidence that immunomodulatory Lactobacilli affect the proliferation of the new coronavirus [26].

## 6. Mixed Trials with Probiotic Administration in COVID-19 Infection

Kageyama et al. combined in vitro experiments and a clinical double-blind prospective study to determine cytokine production and evaluate immune efficacy of lactic acid bacteria against COVID-19. Healthy adults with neither past nor current SARS-CoV-2 infection were enrolled. The clinical trial focused on *L. plantarum*, *Bifidobacterium longum*, and *Lactococcus lactis* ssp. *lactis*, which have strong protective effects against RNA virus infection of the respiratory tract. In in vitro cytokine response assay, immune cells collected from peripheral blood from each participant were treated with each heat-killed probiotic strain to assess probiotics’ ability to stimulate cytokine production. The more and less immunomodulatory potent species matched for each participant were selected, and subjects ingested the live form of probiotic strains.

The intervention lasted three weeks. The best-matched species were administrated on the first week of the trial, then a one-week washout period followed, and on the third week, participants ingested the worst-matched species. Oral administration of *L. plantarum* significantly enhanced the activity of natural killer (NK) cells, achieving a marked increase in nonspecific immune response cytokine levels, while decreasing the levels of IL-6 in blood plasma. In contrast, oral administration of *B. longum* failed to significantly stimulate cytokine production. *L. plantarum* exhibited outstanding immunomodulatory ability, mimicking the inflammatory responses of nonspecific immunity, which is necessary in the early stages of a host’s defense mechanism against viral infection. Using one specific formula to measure anti-COVID-19 immunomodulatory potential of the lactic acid bacteria, results from in vitro stimulation assay confirmed the ability of *L. plantarum* to remarkably increase the nonspecific immune response-induced levels of cytokines in all subjects’ samples [27].

## 7. Discussion

Currently, preclinical studies provide evidence that Lactobacilli strains—*L. rhamnosus* and *L. Plantarum*—pose beneficial effects to ameliorate the inflammatory response against SARS-CoV-2 [21,23]. Additionally, probiotics confer specific immune responses in inflammatory cytokine expression, prevent cell apoptosis, and induce immunological memory [22]. At the same time, Mak et al. claimed that “one size does not fit all”, meaning that not all probiotics are alike. Lactobacilli and Bifidobacteria are only two types of nonpathogenic bacteria, while there are more to be studied. The idea of probiotics as a cotreatment for COVID-19 is derived from indirect evidence, while empirical use is not recommended [28].

Hung et al. supported the theory that probiotics may have antiviral effects via the gut–lung axis and could improve gut health by achieving homeostasis. Their review included 19 ongoing trials of probiotics for the prevention or adjuvant therapy of COVID-19 registered in ClinicalTrials.gov until June 2021. The primary outcome was divergent in these trials and varied among disease prevention, symptom relief, antibody titers, disease progression, viral load, effects on gut microbiota, and mortality [29]. Another possible drawback could be that the majority of trials used probiotic mixtures, making it doubtful to identify the potentially beneficial strain or the optimal CFUs (colony-forming units). A meta-analysis of completed clinical trials supported that the duration of respiratory illness episodes is significantly reduced with probiotic treatment compared to placebo [30]. On the contrary, Hao et al. reported moderate efficacy of probiotics for preventing acute upper respiratory tract infections [31]. Wu et al. showed that in patients with pneumonia from COVID-19, the treatment with simultaneous intake of nine probiotic strains of Lactobacilli led to improved clinical outcomes and lower levels of inflammatory biomarkers (TNF-α, interleukin IL-1β) [23]. Another study with positive results reported that probiotic administration achieved complete remission in 53.1% of patients versus 28.1% in the placebo group. Probiotics reduced viral load in the nasopharyngeal cavity, pulmonary infiltrate, and duration of gastrointestinal and nongastrointestinal symptoms compared to placebo. Remarkably, probiotics significantly increased the production of specific IgG and IgM antibodies against SARS-CoV2 compared to placebo, confirming their immunomodulatory effect [24]. Mullish et al. also showed a reduction of 27% in viral upper respiratory tract infection symptoms in obese and overweight subjects compared to the control group after Lactobacillus and Bifidobacteria administration. Subjects over 45 years or with body mass index higher than 30 kg/m^2^ showed a significant reduction, indicating positive findings potentially usable in patients with COVID-19 disease [32].

During the COVID-19 pandemic, 2.3% of affected patients underwent invasive mechanical ventilation [33]. Τwo randomized controlled clinical trials showed that probiotic strains of *L. rhamnosus* GG, *Bacillus subtilis*, and *Enterococcus faecalis* administration in mechanically ventilated, critically ill patients was associated with lower incidence and less severe ventilator-associated pneumonia compared to the control group [34,35]. However, the efficacy of probiotics in reducing mortality rates in ICU and hospitalized patients was considered uncertain according to Mak et al. [28]. On the other hand, D’Ettorre et al. supported that probiotic administration was associated with significant reduction of the estimated risk of respiratory failure, need for ICU hospitalization, and mortality risk [5].

Gut microbiota alterations and translocation of pathobionts regulate various immunometabolic activities, and they are associated with susceptibility to sepsis. For instance, D-lactate, an essential enzyme for the integrity of the Kupffer cells, is produced from gut bacteria. Kupffer cells, which translocate to liver through the portal vein, capture and kill circulating pathogens. Butyrate acid mediates macrophage metabolism through monocyte transformation into macrophages and histone deacetylase inhibition, enhancing antimicrobial agents’ production [36]. According to Kumar et al., probiotics should be administered in septic patients with caution, especially in neonates and the elderly, as dysregulation of the gut flora could exacerbate sepsis by stimulating the production of proinflammatory cytokines [37]. Albeit, Shimizu et al. supported that administration of *Bifidobacterium breve* and *Lactobacillus casei* increased the levels of beneficial bacteria and SCFAs in the gut, highlighting that their prophylactic use could potentially prevent the incidence of enteritis and VAP in septic patients, through the modification of intestinal flora [38]. It is undeniable that probiotics have strain- and dose-specific properties; therefore, their effects should not be generalized.

## 8. Conclusions

The immunomodulatory action of probiotics through the gut–lung axis affects inflammatory biomarkers and the microbial flora of lungs, classifying them as a potential complementary therapeutic weapon against SARS-CoV-2.

However, the reviewed preclinical and clinical trials suffer from design flaws and different outcome measures. Heterogeneity was observed in the use of bacterial strains and populations, with a preference in the probiotic mixtures, in experimental and clinical studies as well as in the target groups. In vitro trials with single strain probiotic *L. plantarum* showed promising results against SARS-CoV-2 [39]. As shown in Figure 4, in future research, it would be helpful to repeat experiments with this single strain probiotic or the bacteriocin derived from *L. plantarum*, namely, Plantaricin, in epithelial cell lines (alveolar and intestinal) inoculated with SARS-CoV-2 to detect biomarkers (like IL-6) associated with beneficial probiotic effects.

These biomarkers should be confirmed by experimental studies using an animal model, and thereafter the schedule continues with dose-finding phase II clinical trials. Finally, efficacy and safety of the intervention, as well as biomarkers used in clinical practice must be confirmed in phase III clinical trials to establish probiotics as cotreatment against SARS-CoV-2 infection.

## Figures and Tables

**Figure 1 microorganisms-10-01764-f001:**
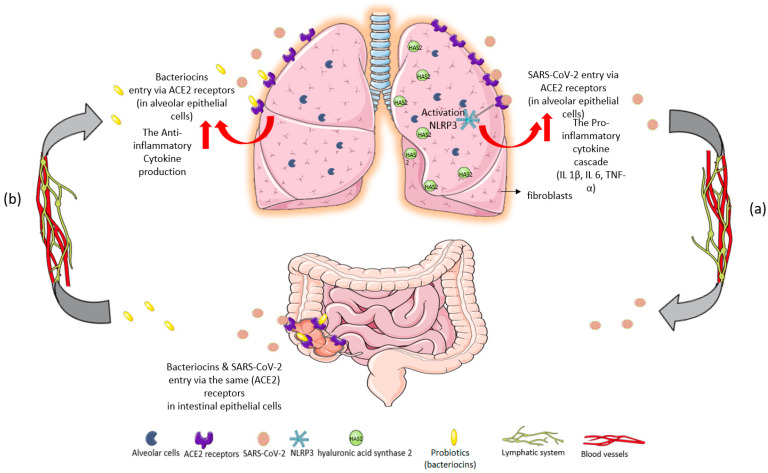
Immunomodulatory response of gut–lung axis cross-talk after: (**a**) SARS-CoV-2 invasion and (**b**) probiotic intake of *L. plantarum*. (**a**) SARS-CoV-2 binds to ACE2 receptors in the epithelial cells of the lungs. The massive cytokine release after viral entry leads to NOD-, LRR-, and pyrin domain-containing protein 3 (NLRP3) inflammasome overactivation, exacerbating the inflammatory cascade [9]. High levels of proinflammatory cytokines IL-1β, IL-6, and TNF-α are inducers of hyaluronic acid synthase 2 (HAS2) in the endothelium, alveolar epithelial cells, and fibroblasts. Hyperinflammation leads to hyaluronic acid production and accumulation in the alveoli of patients with severe COVID-19. The development of pulmonary fibrosis occurs due to inflammatory response and severe infection. The virus infects intestinal epithelial cells in the same way that it directly attacks alveolar epithelial cells. (**b**) Probiotics could restrict SARS-CoV-2 invasion and proliferation through their metabolites, bacteriocins (antimicrobial peptides), because of their ability to block ACE2 receptors [9]. Specifically, *L. plantarum* bacteriocins compete with the virus for the same host sites [10]. Thus, probiotics could offer a protective effect against acute respiratory distress syndrome (ARDS) in COVID-19 infection, reverse the “cytokine storm”, and reduce hyaluronic acid synthesis in the lungs [11,12].

**Figure 2 microorganisms-10-01764-f002:**
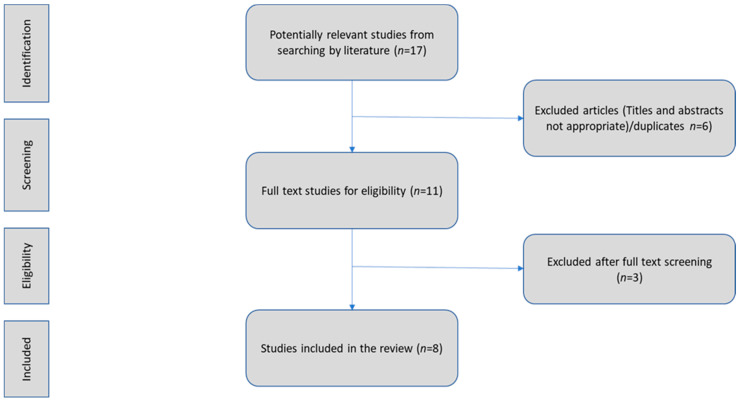
Flow chart. Identification and selection of the studies.

**Figure 3 microorganisms-10-01764-f003:**
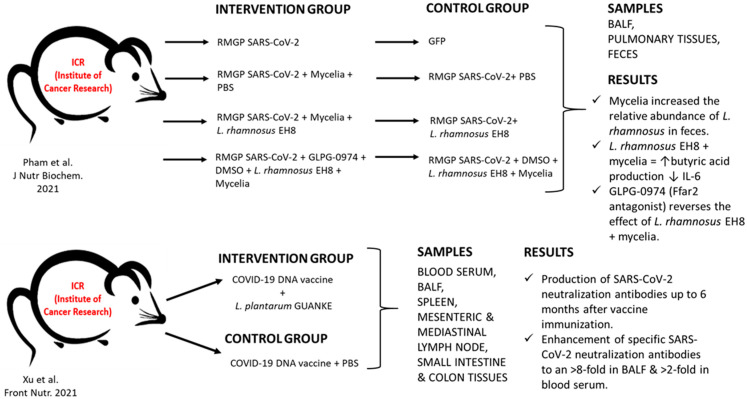
Preclinical studies of probiotic administration in COVID-19 infection [21,22]. RMGP: Recombinant membrane glycoprotein; GFP: green fluorescent protein; PBS: phosphate buffered saline; *L. rhamnosus* EH8: *Lactobacillus rhamnosus* strain EH8; DMSO: Dimethyl sulfoxide; GLPG-0974: 4-[[(R)-1-(Benzo[b]thiophene-3-carbonyl)-2-methyl-azetidine-2-carbonyl]-(3-chloro-benzyl)-amino]-butyric acid, Ffar2: Free fatty acid receptor 2; BALF: Bronchoalveolar lavage fluid.

**Figure 4 microorganisms-10-01764-f004:**
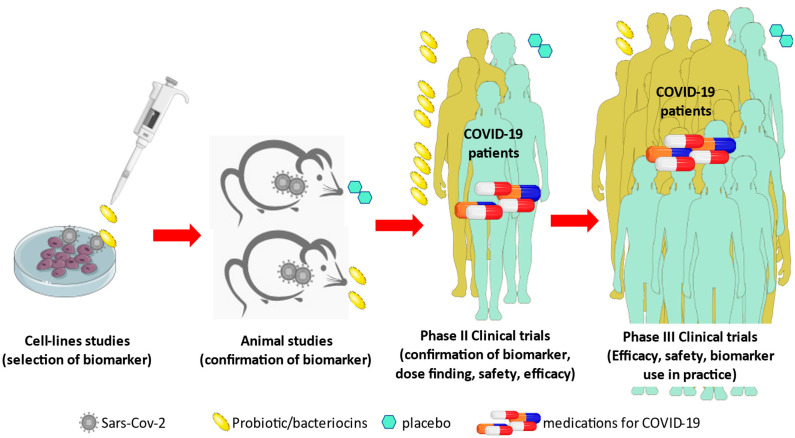
Proposed schedule of studies and their outcome measures to establish probiotic administration as supplementary treatment of COVID-19.

**Table 1 microorganisms-10-01764-t001:** Clinical trials with probiotic administration in COVID-19 infection.

Clinical Trials with Probiotic Administration in COVID-19 Infection						
Reference	Study Group (SG)	Intervention	Control Group (CG)	Intervention	Biological Samples	Results
Wu et al. [23]	COVID-19 patients (n = 13)	*B. lactis* HNO19 *L. casei* Lc-11 *L. plantarum* Lp-15 *B. lactis* B420 *B. longum* BL05 *L. format* Lg-36 *L. rhamnosus* Lr-32 *L. paracasei* Lpc-37 *L. salivarius*	Patients with community-acquired pneumonia (n = 15)	*B. lactis* HNO19 *L. casei* Lc-11 *L. plantarum* Lp-15 *B. lactis* B420 *B. longum* BL05 *L. format* Lg-36 *L. rhamnosus* Lr-32 *L. paracasei* Lpc-37 *L. salivarius*	Feces	Restoration of intestinal dysbiosis and pulmonary dysfunction. Reduced inflammatory biomarkers: ↓TNF-α, ↓ IL-1β, ↓IL-4, and ↓IL-12P70.
			Healthy controls (n = 15)	---		
Gutiérrez-Castrellón et al. (RCT) [24]	COVID-19 ICU patients (n = 150)	*L. plantarum* KABP022 *L. plantarum* KABP023 *L. plantarum* KABP033 *P. acidilactici* KABP021	COVID-19 ICU patients (n = 150)	Placebo	Nasopharyngeal specimens, blood, and feces	Complete remission in 53.1% of patients in SG and 28.1% in CG. Reduced viral load, lung infiltrates, and symptoms duration in SG. Significant increase in IgM and IgG against SARS-CoV-2 and faster reduction of D-Dimers in SG.
D’Ettorre et al. [5]	COVID-19 patients (n = 28)	*S. thermophilus* DSM 32345 *L. acidophilus* DSM 32241 *L. helveticus* DSM 32242 *L. paracasei* DSM 32243 *L. plantarum* DSM 32244 *L. brevis* DSM 27961 *B. lactis* DSM 32246 *B. lactis* DSM 32247	COVID-19 patients (n = 42)	---	---	SG: remission of diarrhea and other symptoms 72 h after oral bacteriotherapy. Estimated risk of developing respiratory failure: eight-fold lower in SG. CG: Higher ICU admission and mortality rates.
Fernández-Ferreiro et al. (RCT) [25]	Elderly people vaccinated with mRNA-based vaccine against SARS-CoV-2 (n = 98)	*L. coryniformis* K8 CECT 5711	Elderly people vaccinated with mRNA-based vaccine against SARS-CoV-2 (n = 100)	Placebo	Blood	*L. coryniformis* K8: enhances vaccine-specific immune responses against SARS-CoV-2 in elderly populations. ↑IgG, ↑IgA in SG.

## Data Availability

Not applicable.

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
