# Peer review of "Immunomodulatory Effects of Probiotics on COVID-19 Infection by Targeting the Gut–Lung Axis Microbial Cross-Talk"

_microorganisms, 2022, doi:10.3390/microorganisms10091764_

Round 1

Reviewer 1 Report

The article is consistent within itself. It successfully explained why current research is important. Nevertheless, it stated the purpose of the research question and provide discussion over them.

The references are relevant and recent. The cited sources are referenced correctly. Appropriate and key studies are included. The paper is comprehensive, the flow is logical, and the data is presented critically.

Illustrative materials (figures and tables) are clear, comprehensive and very informative.

However, there are some specific comments on weaknesses of the article and what could be improved:

Specific comments on weaknesses of the article and what could be improved:

Major points - none

Minor points

1. Paragraphs are too abundant to follow easily. some of the main ideas should be transferred to a new paragraph.

2. Please, organize the main recommendations based on the literature review, 

Author Response

Response to Reviewer 1 Comments

  1. Paragraphs are too abundant to follow easily. some of the main ideas should be transferred to a new paragraph.

Response: Each paragraph analyzes one specific study, so the extent of the paragraph is proportional to the survey that is cited.

  1. Please, organize the main recommendations based on the literature review, 

Response:  In accordance with our review, there is no yet specific recommendations for the use of probiotics (strain or CFUs) during the infection from SARS-COV-2.  There are some surveys that support the antiviral effects of probiotics but further research is needed.  However, the heterogeneity that is observed in the use of bacterial strains and populations, with a preference in the probiotic mixtures, in experimental and clinical studies, as well as in the target groups can not lead to safe conclusions.

Reviewer 2 Report

The authors have performed a literature review with the aim to explore the effect of probiotics on the immune response to COVID-19 with focus on the gut-lung axis microbial crosstalk. The topic is of general interest to the field and the manuscript is well-written, however, the total number of references included in the review is very low. The study could be improved by adding more background evidence to preclinical and clinical studies “confirming” the gut-lung axis communication in COVID-19.

Major comments:

-          The literature search strategy seems reasonable and to the point, however, comes across as a bit strict and may overlook several relevant studies. This could be mentioned as a possible limitation of the paper. Of note, a recent review paper assessed 19 registered clinical trials at ClinicalTrials.gov (many of which are ongoing) investigating the potential effect of probiotics in COVID-19 (Hung et al 2021, https://doi.org/10.3390/microorganisms9081605). The authors could consider mentioning this paper and/or discuss the included publications in light of this information, as it could serve as an extension to the work.

-          The references included in the background information (section 1.1 and 1.2) are very few and should preferably include more to convince the reader of the rationale behind the work. Moreover, the wording “confirming” is a bit unfortunate, I would rather use “supporting” or similar.

-          Likewise to the point above, sections 5 and 6 (In vitro trials and mixed trials) only include one reference each. Is this really the case that there are no more studies found?

Minor comments:

-          In the section that mentions reference nr 7 (Vestad et al), there are a couple of mistakes in results reporting:

o   Line 128 should be: Results showed that----in COVID-19 patients with respiratory dysfunction (not failure, as they define acute resp failure and persistent dysfunction differently)----

o   Line 138: Respiratory dysfunction (not failure)

-          Be aware that reporting of some bacterial taxa should be in italics, such as genus level, i.e Veillonella.

-          In section 4, the two first studies mentioned (ref 17 and 18) are perhaps lacking an introductory sentence (Wu et al investigated---- etc?)

-          Several of the references are not consistently reported (some are lacking year etc).

Author Response

Response to Reviewer 2 Comments

Major comments:

  • The literature search strategy seems reasonable and to the point, however, comes across as a bit strict and may overlook several relevant studies. This could be mentioned as a possible limitation of the paper. Of note, a recent review paper assessed 19 registered clinical trials at ClinicalTrials.gov (many of which are ongoing) investigating the potential effect of probiotics in COVID-19 (Hung et al 2021, https://doi.org/10.3390/microorganisms9081605). The authors could consider mentioning this paper and/or discuss the included publications in light of this information, as it could serve as an extension to the work.

Response: Thank you for the remark; We made the necessary changes to fulfil your request and added a new paragraph which describes the work of Hung and his colleagues about the ongoing trials of probiotics (line 380-388)

  • The references included in the background information (section 1.1 and 1.2) are very few and should preferably include more to convince the reader of the rationale behind the work. Moreover, the wording “confirming” is a bit unfortunate, I would rather use “supporting” or similar.

Response: Thank you for the suggestion; Done as requested. New paragraphs added in both sections (line 104-121 & 124-166) and the grammatical errors have been changed.

  • Likewise to the point above, sections 5 and 6 (In vitro trials and mixed trials) only include one reference each. Is this really the case that there are no more studies found?

Response: To the best of our knowledge, other studies that fulfil our criteria do not exist.

Minor comments:

-          In the section that mentions reference nr 7 (Vestad et al), there are a couple of mistakes in results reporting:

o   Line 128 should be: Results showed that----in COVID-19 patients with respiratory dysfunction (not failure, as they define acute resp failure and persistent dysfunction differently)----

o   Line 138: Respiratory dysfunction (not failure)

Response:  Done, as requested

-          Be aware that reporting of some bacterial taxa should be in italics, such as genus level, i.e Veillonella.

Response:  We made all the changes, as you have suggested. In this case we would like to inform you that in the first manuscript that we submitted, bacterial taxa were in italics. Probably, there is a problem with the system and make changes automatically. This may cause inconvenience to other authors also

-          In section 4, the two first studies mentioned (ref 17 and 18) are perhaps lacking an introductory sentence (Wu et al investigated---- etc?)

Response: Corrected

-          Several of the references are not consistently reported (some are lacking year etc).

Response: Thank you for the remark; Done, as requested

Round 2

Reviewer 2 Report

Dear authors,

Thank you for addressing these issues. In general, a literature review should preferably contain even more references, but due to the narrow focus and that the field is quite new, I find it reasonable to recommend your article for acceptation and I am happy with the changes made. Best of luck!